# Subtrochanteric Insufficiency Fracture Occurring 5 Years after Surgery at the Steinmann Pin Insertion Site for Fracture Reduction

**DOI:** 10.3390/medicina58030404

**Published:** 2022-03-09

**Authors:** Chang-Hwa Hong, Jong-Seok Park, Byung-Woong Jang, Heejun Jang, Chang-Hyun Kim

**Affiliations:** 1Department of Orthopedic Surgery, Soonchunhyang University Hospital Cheonan, 31, Suncheonhyang 6-gil, Dongnam-gu, Cheonan 31151, Chungcheongnam-do, Korea; chhong@schmc.ac.kr (C.-H.H.); jsparksch@schmc.ac.kr (J.-S.P.); 129651@schmc.ac.kr (H.J.); 2Department of Orthopedic Surgery, Soonchunhyang University Hospital Gumi, 179, 1 Gongdan-ro, Gumi 39371, Gyeongsangbuk-do, Korea; 99845@schmc.ac.kr

**Keywords:** femur, bone wires, osteonecrosis, insufficiency fractures, fracture reduction

## Abstract

*Background and Objectives*: Steinmann pins are commonly used in orthopedics, with a low rate of complications. However, thermal osteonecrosis may occur when a pin is inserted using a drill. There have been no reports on late-onset fractures at the Steinmann pin insertion site. *Materials and Methods*: A 32-year-old man who underwent surgery for a femoral shaft fracture 5 years ago complained of proximal thigh pain 1 month after the removal of the internal device. On physical examination, the patient showed a limping gait due to pain, and tenderness was observed on the lateral aspect of the proximal thigh. Magnetic resonance imaging was performed because the symptoms did not improve, despite conservative treatment. A new fracture line was observed in the lateral cortical bone of the proximal femur. It was found that a fracture occurred at the site where the Steinmann pin was inserted for a closed reduction at the time of the first operation. The patient was instructed to limit weight bearing and to use crutches while walking. Parathyroid hormone was additionally administered to promote bone formation. *Results*: Six months after diagnosis, a complete union was achieved at the subtrochanteric fracture site, and the patient’s pain subsided. *Conclusions*: A fracture that occurs as a late onset at the provisional Steinmann pin insertion site is an extremely rare complication; however, orthopedic surgeons must consider this possibility and make more efforts to lower the occurrence of thermal damage. In addition, if the patient complains of pain in the region where the pin was inserted after surgery, surgeons should spare no effort to determine whether a new fracture has occurred.

## 1. Introduction

Kirschner wires and Steinmann pins are commonly used in orthopedic surgery. In fracture surgery, Kirschner wires and Steinmann pins facilitate fracture reduction and are highly effective as provisional fixations to maintain reduction. Especially in children, these are the most used tools for definitive fracture fixation. These pins are considered safe and effective, with a low incidence of complications. However, thermal osteonecrosis may occur when a pin is inserted using a drill [1,2]. As Steinmann pins have a very small insertion part, thermal damage to the bone heals itself without causing clinical complications in most cases. In our review of the literature, there are no reports of late-onset fractures at the Steinmann pin insertion site. We present a case of subtrochanteric insufficiency fracture that occurred 5 years after surgery at the site where the Steinmann pin was inserted for fracture reduction.

## 2. Materials and Methods

A 32-year-old man was admitted to the emergency department after being injured in a traffic accident. **He had no underlying disease or relevant medical history.** The patient was diagnosed with a comminuted fracture of the right femoral shaft and underwent fixation surgery using an intramedullary nail. At the time of surgery, for a closed reduction of the fracture site, a Steinmann pin was inserted 1 cm below the lesser trochanter, and fracture reduction was attempted by the “joystick technique”. After the nail was inserted, the Steinmann pin was removed. Sixteen months after surgery, autogenous bone grafting was performed for nonunion of the fracture site, and the intramedullary nail was removed and fixed with a plate. Fracture site union was achieved 1 year after the reoperation. **The patient was young, and there was no pain, but he complained of discomfort during physical activity, so he wanted to remove the internal device.** Therefore, five years after the first operation, he underwent surgery for plate removal.

One month after the final surgery, the patient complained of pain in the right proximal thigh, which was not present before the surgery. On physical examination, the patient showed a limping gait due to pain, and tenderness was observed on the lateral aspect of the proximal thigh. **Blood tests, including C-reactive protein, were all within the normal range.** Since no findings were observed on conventional radiography and it was not long after the surgery, we decided to follow up on the change in symptoms. Two months after surgery, the patient complained of persistent unexplained localized proximal thigh pain, and magnetic resonance imaging (MRI) was performed. As the MRI scan showed a linear signal intensity change in the lateral cortex of the subtrochanteric region, it was established that a new linear fracture may have occurred (Figure 1). Since the subtrochanteric fracture did not occur at the site where the screw was removed, the images taken from the first visit to the emergency room were serially reviewed to determine the cause. Surprisingly, it was found that a fracture occurred at the site where the Steinmann pin was inserted for closed reduction at the time of the first operation.

As the fracture line suspected on MRI was incomplete and stable, the patient was instructed to limit weight bearing and to use crutches while walking. Conventional radiography performed 4 months after the final operation revealed an incomplete transverse fracture, bone resorption and callus formation on the lateral cortex (Figure 2).

Parathyroid hormone (PTH) was additionally administered to promote bone formation. After 6 months, the subtrochanteric fracture site achieved complete union, and the patient’s pain subsided (Figure 3).

## 3. Discussion

This case suggests that an insufficiency fracture can occur at the site where the Steinmann pin was inserted even after a long period of time. Steinmann pins have been used in orthopedics for provisional fixation, percutaneous fixation, and as reduction tools during fracture surgery [3,4,5,6]. Steinmann pins are commonly used as tools to reduce long bone fractures [7]. In particular, it is quite effective to insert a Steinmann pin or a Schanz screw and use it as a joystick for the indirect reduction in the proximal part during proximal femoral fracture surgery.

There are few complications associated with the use of Kirschner wires or Steinmann pins. The main complications that occur often serve as the reason for the definitive fixation of fractures. The inserted pin may migrate, or pin tract infection may occur after percutaneous pinning [8,9]. The migrated pin is resolved by removing it, and because pin tract infection is usually a superficial infection, good results can be obtained if properly treated. Another complication of percutaneous pinning is that the soft tissue cannot be directly protected; thus, peripheral nerve damage may occur [10]. Finally, as in this case, thermal damage when inserting a Steinmann pin using a drill can cause osteonecrosis in normal bone [11]. In a cadaver study, Matthews et al. [12] reported that heat exceeding 55 °C was generated during pin insertion using a drill. However, there have been no reports of fractures occurring after a long period due to osteonecrosis. We searched the electronic database, PubMed, to find similar case reports. We searched the terms “(fracture, femur, Steinmann pin, closed reduction, joystick technique) AND (osteonecrosis, a late-onset fracture, pin site complication)”. This is a rare case where the Steinmann pin inserted for the joystick technique reduction did not bring about healing, and a fracture occurred 5 years after the operation. Although rare, it is quite important because there is a problem that additional efforts, such as surgical treatment for new fractures, are required when all treatments are completed, and the patient also experiences problems, such as an additional treatment period and the burden of treatment costs. In addition, as in this case, when a fracture occurs in long bones, such as the subtrochanteric region, it is necessary to recognize that there is a risk of new complications, such as massive bleeding, pulmonary embolisms and unexpected permanent sequelae.

It was unforeseen that a new fracture occurred at the site where a Steinmann pin was inserted at the time of surgery. Figure 4 shows serial radiography from the first operation to the time of the discovery of the subtrochanteric fracture (Figure 4). An unhealed hole was also observed on computed tomography (CT) scan taken 4 years after the first operation (Figure 5). The traces of the Steinmann pin insertion visible after the first operation remained even after 5 years. It is thought that when the plate, which distributes the load, is removed, the pin insertion hole acts as a stress bearer, and an insufficiency fracture occurs.

Although a new fracture line was suspected on MRI, it was challenging to determine whether it was a clear fracture because it was masked by the traces remaining after the screws were removed. If actions such as restrictions on ambulation were not taken, it would likely have progressed to a complete fracture.

This case can be regarded as an insufficiency fracture because it occurred without major trauma to the abnormal bone damaged by thermal necrosis. Additionally, this fracture also has the features of an insufficiency fracture, as defined by an American Society for Bone and Mineral Research task force [12]: (1) The fracture is located in the subtrochanteric area. (2) The fracture was not associated with trauma. (3) It was a noncomminuted fracture and had a transverse configuration. (4) It was an incomplete fracture involving only the lateral cortex and showed a localized periosteal reaction of the lateral cortex.

Several recommendations have been made to prevent osteonecrosis due to thermal damage. However, in practice, these preventive methods often have limitations in their application. As thermal damage mainly occurs at high speeds, the pin should be inserted at low speeds and with frequent drill stops [13]. However, since the cortex of the femur in a young and healthy male patient is extremely hard, inserting the Steinmann pin at a low speed requires more time and patience from the orthopedic surgeon. A study on knee arthroplasty reported that irrigation could significantly reduce the thermal damage caused by burring and sawing [14]. However, this method cannot be applied during percutaneous pinning because the part of the bone where the pin is inserted will not be exposed.

As aforementioned, methods to reduce thermal damage can require time, patience, and in some cases, can be challenging to apply. In addition, even if osteonecrosis occurs in the pin insertion part, if it is not a small bone such as the phalanx, it rarely affects the clinical course, so efforts to prevent it may be neglected.

In this case, there was a risk that a small Steinmann pinhole could completely change the patient’s clinical course. Even if surgical treatment was not necessary because it did not progress to a complete fracture, the patient had to use crutches for a long time, even after the removal of the plate, had to limit weight bearing while walking, and suffered the inconvenience of increased medical expenses as PTH was administered.

In this case, PTH was administered to promote the healing of fractures that occurred in the area that had been sclerotic for a long time. This is because the net anabolic effect on bone of PTH can enhance fracture healing [15]. In this situation, other methods may be chosen to enhance fracture healing. Extracorporeal shock waves (ESWT) induce differentiation and the proliferation of stem cells and increase osteoblastic activity, which can promote the regeneration of fracture sites [16]. In addition, pulsed electromagnetic field (PEMF), which has been approved by the Food and Drug Administration as a method for the treatment of fracture nonunion and promotion of bone formation, is also a good option [17].

When inserting a Steinmann pin, even if temporarily, it should be kept in mind that osteonecrosis caused by thermal damage may remain unresolved for a longer time than expected and may lead to complications. It is advisable to engage in efforts to reduce the damage as much as possible during surgery. Furthermore, when a patient complains of pain in an unexplained area after surgery, especially if the area is a provisional pin insertion site during surgery, it is necessary to pay attention to whether a new fracture has possibly occurred.

This report has several limitations. First, as no reports similar to this case could be found, literature references were limited. Second, it is unknown how much the damaged biology caused by the change of fixation device contributed to the failure of the pin insertion site to heal. Moreover, and importantly, because the lesion was seen on CT before surgery to remove the internal device, sufficient warnings should have been provided to the patient before surgery.

## 4. Conclusions

In conclusion, a late-onset fracture that occurs at the provisional Steinmann pin insertion site is a rare complication; however, orthopedic surgeons must consider this possibility and make more efforts to reduce the occurrence of thermal damage. In addition, if the patient complains of pain in the region where the pin was inserted after surgery, surgeons should spare no effort to determine whether a new fracture has occurred.

## Figures and Tables

**Figure 1 medicina-58-00404-f001:**
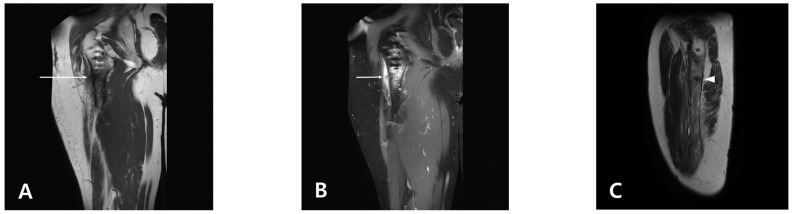
Magnetic resonance imaging (MRI) of the proximal thigh. (**A**) Linear signal intensity change in the lateral cortex of the subtrochanteric region in the T1-weighted image (white arrow); (**B**) T2-weighted STIR image revealed low signal intensity in the lateral cortex of the femur with trabecular edema (white arrow); (**C**) Sagittal image revealed focal low signal intensity in the T2-weighted image (white arrowhead).

**Figure 2 medicina-58-00404-f002:**
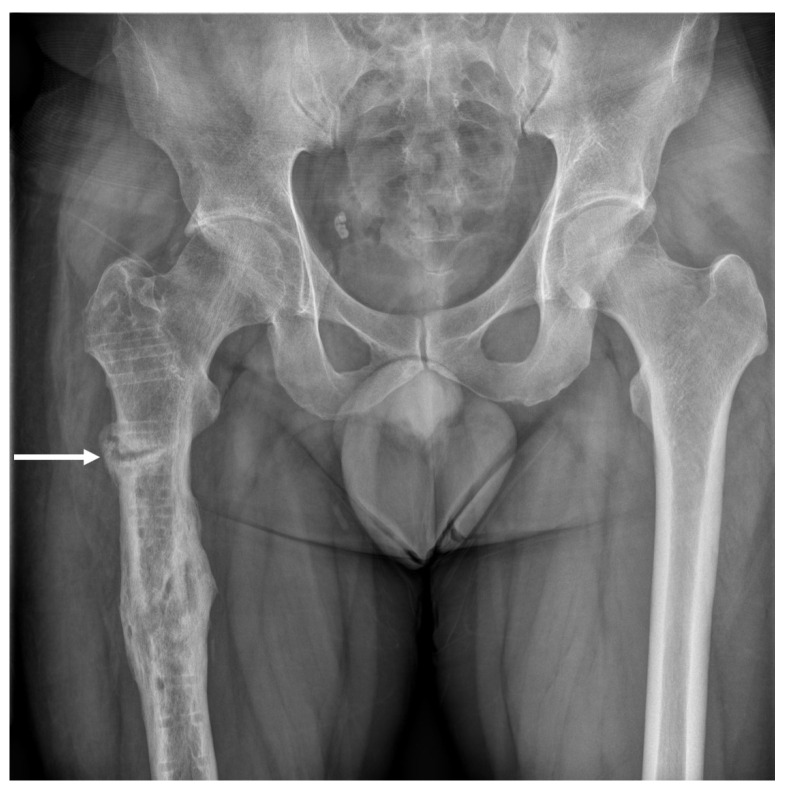
Conventional radiography was performed 4 months after the final operation. The white arrow indicates an incomplete transverse fracture in the lateral cortex of the right femur.

**Figure 3 medicina-58-00404-f003:**
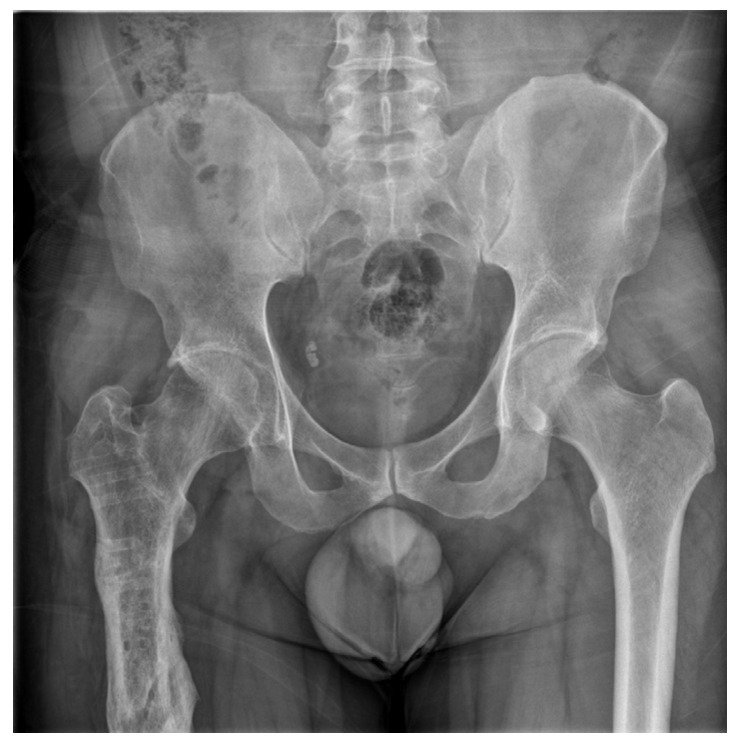
Conventional radiography was performed 6 months after the final operation. The fracture site achieved complete union.

**Figure 4 medicina-58-00404-f004:**
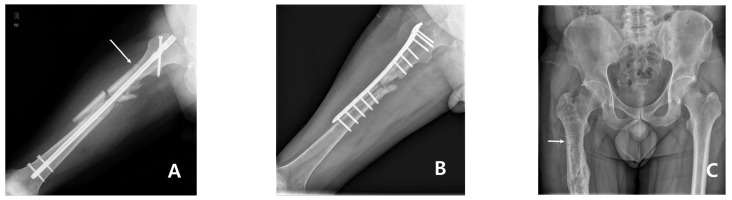
Serial conventional radiography from the first operation to the time of the discovery of the fracture. (**A**) The traces of the Steinmann pin insertion visible after the first operation; (**B**) Six months later, after internal device replacement surgery, trace is visible between the screws; (**C**) The trace remained even after 5 years from first surgery (white arrow).

**Figure 5 medicina-58-00404-f005:**
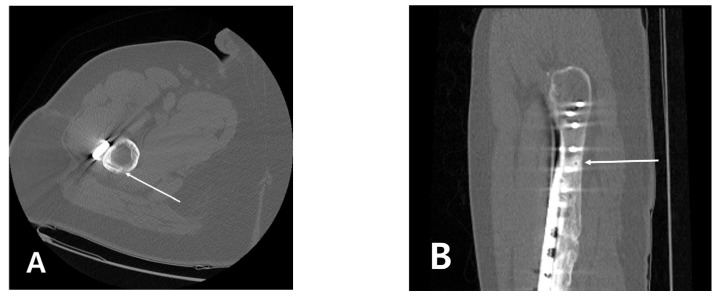
Computed tomography (CT) scan taken 4 years after the first operation. (**A**) The axial image showed linear trace along the cortex; (**B**) The sagittal image also revealed an unhealed pinhole in the proximal femur (white arrow).

## Data Availability

Not applicable.

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
