# Peer review of "Subtrochanteric Insufficiency Fracture Occurring 5 Years after Surgery at the Steinmann Pin Insertion Site for Fracture Reduction"

_medicina, 2022, doi:10.3390/medicina58030404_

Round 1

Reviewer 1 Report

This is case report Case Report. a fracture occurred at the site where the Steinmann pin was inserted for closed reduction at the time of the first operation.

Line 46: add the past history.

Line 57: add the blood test.

Figure 1: Is it T1 or T2?.  Add the other plane, because Figure 1 is main figure.

Figure 4: please place the figure number according to the time flow.

Line 157: How rare is “extremely rare” ? please add the literature review on this similar cases.

Author Response

Thank you for your careful review and kind comment.

Thank you in advance.

Reviewer 2 Report

Dear authors!

I liked reading your manuscript.

It covers a theme of a very seldom complication, however a serious one.

Minor spell check required

It is relevant and interesting. It is a case report, therefore data is very limited. Fractures after Steinmann pinning are rare therefore litte data is existing. The text is clear and easy to read. The conclusions are consistent with the evidence and arguments presented

With kind regards

Author Response

(The authors gave the same response as above.)

Reviewer 3 Report

Dear Authors,

as regards the M&M, you have to describe why you propose the removal of the plate after 5 years. In detail you have to specify the clinical and to add an x rays in order to justify the removal surgery.

Furthermore as regards the MRI it is usefull to add a stir view in order to evidence the signal of frature and intensity of trabecular oedema.

At the end it is necessary to specify the study protocol of your review: the word that you use in order to research the articles, which database you used, etc….

It is necessary to describe your results because there is not a results section.

As regards the Discussion,

it is difficulty to demonstrate the direct relationship between pin hole and rifracture only by xrays. It will be usefull a CT scan in order to describe better the relationship or to show the screws holes.

In fact it is reasonable to think that the fracture due to the screws removal and the lack of the plate associate also to pin hole.

For this reason it will be interesting for the reader to see CT scan and the Authors will be underline the possible relationship betwen implant removal, pin hole and risk of fracture.

Moreover the Authors will have to improve the part of therapy regarding the actual strategy including drug therapy (PTH etc …..) and shockwavetherapy which represent a modern treatment for fracture and re-fracture.

You could cite the following article in which the Authors describe the foundamental principle of shockwave and fracture healing.

-Extracorporeal shock waves induce osteogenic differentiation of human bone-marrow stromal cells

Journal of Biological Regulators and Homeostatic AgentsVolume 30, Issue 4, Pages 139 - 144September-December 2016

Notarnicola A,Vicenti G,Maccagnano G,Silvestis F,Cafforio P,Moretti B

-The role of biophysical stimulation with pemfs in fracture healing: From bench to bedside

Journal of Biological Regulators and Homeostatic AgentsVolume 34, Issue 5, Pages 131 - 135September-October 2020

Vicenti G, Bizzoca D, Solarino G ,Moretti F, Ottaviani G, Simone F, Zavattini G, Maccagnano G, Noia G, Moretti B

As regards the conclusion you have to improve in relation to the modifications of the article

Author Response

(The authors gave the same response as above.)

Round 2

Reviewer 1 Report

Thank you for revision.

Examination: add the objective data to diagnosis the fracture as “insufficiency fracture “.

Line 97: at first paragraph, describe the summary of the case.

Line 180: add the limitation in your case report.

Author Response

(The authors gave the same response as above.)

Reviewer 3 Report

Dear Authors,

your modifications are right and you followed my suggestions

Author Response

(The authors gave the same response as above.)
